# Optimizing reliability of RBC signal oscillation measures from hyperpolarized 129Xe MRI

Ivina Mali[1], Bradie Frizzell[1], Steven Haworth[1], Peter J. Niedbalski[1,2]*

1 Division of Pulmonary, Critical Care, and Sleep Medicine, University of Kansas Medical Center, Kansas City, Kansas, United States of America, 2 Hoglund Biomedical Imaging Center, University of Kansas Medical Center, Kansas City, Kansas, United States of America

* pniedbalski@kumc.edu

## Abstract

Recent advances to gas exchange hyperpolarized 129Xe MRI (Xe-MRI) have demonstrated that cardiogenic oscillations within the xenon red blood cell (RBC) signal are sensitive to pulmonary disease. Moreover, by implementing keyhole image reconstruction with gas exchange images collected using standard methodology, maps of regional oscillation amplitude can be generated. While such mapping has been demonstrated on a limited basis, validating these maps remains challenging due to the absence of easily measured biomarkers of pulmonary microvascular health. Moreover, as this is a very new technique, each of the previous implementations has used different methodology; it is unclear which of these methods provides optimal measures of regional oscillation amplitude. In this study, we evaluated oscillation mapping using the different published methods to determine which has the best same-day reliability. Because there are no easily obtainable measures of pulmonary vascular health, reliability serves as a valuable endpoint for validating that these maps are sensitive to real pulmonary physiology. We evaluated the same-day reliability of RBC oscillation measures in patients with systemic sclerosis (N = 6) and pulmonary arterial hypertension (PAH; N = 10), using single-time-point data from healthy volunteers (N = 9) to demonstrate "healthy" oscillation amplitude maps. Global measures of RBC oscillation amplitude (intraclass correlation coefficient; ICC = 0.88) had comparable reliability to standard xenon MRI measures (ICC ≥ 0.82). When examining oscillation mapping, some regional features showed disagreement across scans, but reliability of overall means was strong (ICC ≥ 0.86). Moreover, we show that recent advances in oscillation amplitude mapping for generating both amplitude and phase can provide equivalent maps to those methods that only provide amplitude. Overall, our findings demonstrate that Xe-MRI oscillation mapping has strong reliability when using optimized methods, even in participants with pulmonary disease.

**Data availability statement:** All data used in the production of this manuscript are available from the Harvard Dataverse: https://doi.org/10.7910/DVN/SXAAWQ

**Funding:** This work was supported by the American Heart Association (Career Development Award 930177; PJN), the National Scleroderma Foundation (PJN), the American Lung Association, and the National Institutes of Health (NIH) (R01HL168446; PJN). This study used data from a subset of participants in the Lung Health Cohort, which is supported by the American Lung Association and the NIH (U01HL146408). The funders had no role in study design, data collection and analysis, decision to publish, or preparation of the manuscript.

**Competing interests:** Peter Niedbalski is a consultant for Polarean Imaging, Plc and receives personal consulting fees. Polarean Imaging, Plc had no role in the funding, design, execution, or reporting of this study. This commercial affiliation does not alter our adherence to PLOS ONE policies on sharing data and materials.

## Introduction

Hyperpolarized [129]Xe MRI (Xe-MRI) enables the regional measurement of xenon gas dissolved in red blood cells (RBCs) and other pulmonary tissues ("membrane") [1]. Xe-MRI takes place over a short breath-hold (typically < 15 s), which includes multiple cardiac cycles [2]. As a result, the signal coming from xenon dissolved in RBCs displays cardiogenic oscillations [3,4]. The amplitude of these oscillations has been shown to be sensitive to pulmonary vascular dysfunction [4–6]. In this study, we aimed to optimize the quantification of these oscillations toward facilitating their broader use as a biomarker of pulmonary microvascular function.

Initially, RBC oscillation amplitude was only evaluated on the global level using dynamic spectroscopy. However, standard Xe-MRI gas exchange imaging is collected using center-out radial k-space trajectories, making it amenable to keyhole image reconstruction for creating images of temporal subsets of data [7]. By binning raw data based on oscillation amplitude and using keyhole reconstruction, a regional measurement of RBC oscillation amplitude can be generated. In an initial study, this "oscillation imaging" method showed similar sensitivity to PAH and ILD as global measures [7]. Furthermore, there are cases that appear to demonstrate sensitivity of oscillation imaging to mixed pre-/post capillary pulmonary hypertension and to treatment response in chronic thromboembolic pulmonary hypertension (CTEPH) [8].

Recent developments have suggested that the phase as well as the amplitude of regional RBC signal oscillations can be mapped [9]. This provides further information regarding regional pulmonary hemodynamics. Using this combined amplitude and phase mapping, it was shown that CTEPH patients exhibited greater amplitude heterogeneity and greater regional phase differences as compared to healthy volunteers.

While these initial studies suggest that Xe-MRI oscillation imaging may be an additional effective tool for regional characterization of pulmonary disease, it requires additional optimization and validation before it can be considered a viable screening and monitoring tool. An important step in this process is an assessment of the reliability of the technique, particularly in the PAH population. Suppressed oscillation amplitude appears to be specific to PAH, but this weakened oscillation confounds the binning process that is essential to the oscillation imaging process. This suppressed oscillation amplitude may require the use of different signal processing methods to ensure an accurate identification of oscillation "peaks" and "valleys". To that end, this project examined several possible binning and reconstruction methods in populations of healthy volunteers, PAH patients, and systemic sclerosis patients. This included both the original 2-key approach for mapping oscillation amplitudes and the more recent "multi-key" approach for mapping oscillation amplitude and phase. The effectiveness of binning methods was assessed by examining the same-day reliability of oscillation imaging in participants with pulmonary disease. Upon optimization, the six-week reliability of oscillation imaging was assessed specifically in PAH patients.

## Methods

The basic workflow for generating images of regional oscillation amplitude using the 2-key approach includes binning projections to identify temporal subsets of data with

high RBC and low RBC signal [7]. These subsets are then used to generate two images using keyhole reconstruction: a high key and a low key. The scaled difference between these keys provides a measure of regional oscillation amplitude. In the "multi-key" approach, many key images are generated using a sliding window approach [9]: Starting with the data used to generate the high key from the 2-key approach, a sliding window is used to generate images representing the RBC signal throughout the cardiac cycle. This allows for the evaluation of oscillations within image-space, enabling the measurement of both amplitude and phase within each voxel.

In this work, each of these steps is investigated and optimized to maximize reliability of oscillation imaging measures. Optimization based on reliability was chosen due to a lack of easily measurable clinical correlates. Lung function testing does not directly probe microvascular function: While there may be correlation between Xe-MRI oscillation measures and lung function tests (e.g., forced expiratory volume in 1 second [FEV1], forced vital capacity [FVC], diffusing capacity of the lung for carbon monoxide [DLCO]) [7], these tests probe different physiology than oscillation measures are thought to reflect, namely blood volume oscillations within the pulmonary capillaries. Similarly, CT imaging is unable to resolve the smallest vessels of the lungs and thus does not provide information regarding the pulmonary capillaries. Other MRI methods such as dynamic contrast enhanced (DCE) MRI or phase resolved functional lung (PREFUL) MRI provide a closer measurement, as they measure perfusion based on contrast throughout the cardiac cycle. In contrast, oscillation imaging measures changes to capillary blood volume throughout the cardiac cycle, a more granular measure of capillary function. Right heart catheterization does provide a measure of pulmonary vascular resistance, which is related to oscillation measures [10], but this test is highly invasive and thus unsuitable for the present project. With this lack of viable clinical correlates, reliability serves as a valuable endpoint for optimization.

## Participants

This study evaluated patients with pulmonary arterial hypertension using Xe-MRI. It also included a secondary analysis of data collected for ongoing studies at the University of Kansas Medical Center, including a pilot study in patients with systemic sclerosis and an ancillary study to the Lung Health Cohort [11]. This study was performed with approval from the University of Kansas Medical Center Institutional Review Board (STUDY00146119, STUDY00149906, STUDY00148587) and the FDA (IND 151978). Written informed consent was obtained from all participants. Prospective recruitment of PAH patients began on April 25, 2023, and concluded on November 12, 2024. Deidentified data were accessed for this secondary analysis on December 5, 2024. A total of 25 participant datasets were used for this study: 10 patients with PAH were prospectively recruited for this study. Secondary analysis included 9 young healthy volunteers from the Lung Health Cohort [11] whose images could be used to demonstrate "ideal" healthy oscillation images and 6 patients with systemic sclerosis (SSc)-associated interstitial lung disease (ILD) to test oscillation imaging methods in the case of enhanced oscillation amplitude.

## Gas polarization and delivery

Isotopically enriched (~85% $^{129}$Xe, Polarean Imaging, Durham, NC) xenon gas was polarized using a commercial hyperpolarizer system (9820, Polarean Imaging, Durham, NC). The total dose volume delivered to participants was equal to 20% of forced vital capacity (FVC). Participants received either two xenon doses or three. The first dose was used to collect calibration data according to the $^{129}$Xe MRI Clinical Trials Consortium recommended protocol [2]. This dose contained 20% xenon with the balance ultra-high purity nitrogen. Subsequent doses were used to collect gas exchange images using a single-breath method for collecting high resolution ventilation images and standard gas exchange images following the 1-point Dixon method [12]. Briefly, a 3D spiral acquisition on the gas frequency is interleaved with a 3D radial acquisition centered on the RBC frequency (218 ppm). Essential sequence parameters included: TR = 8 ms (measured between RF pulses on the same contrast), Flip angle = 14.6°, $TR_{90,equiv}$ = 248 ms, TE = 0.45–0.50 ms (such that RBC and Membrane were 90° out of phase), FOV = 400 x 400 x 400 mm$^3$, Gas Matrix = 96 x 96 x 96, Dissolved Matrix = 64 x 64 x 64, Total number

of projections = 1238. Gas exchange imaging doses contained 100% xenon up to 1L. For total dose volumes above 1L, 1L of xenon was used with the balance ultra-high purity nitrogen. In healthy volunteers, a single gas exchange image was collected. In PAH and SSc participants, two gas exchange images were collected in separate breath-holds back-to-back without participants leaving the MRI scanner. PAH patients were additionally imaged using calibration and gas exchange imaging at a second time-point 6 weeks after initial imaging to assess longer-term reliability of oscillation metrics.

## Standard gas exchange analysis

While the present focus is oscillation imaging, it is important to assess the reliability of standard Xe-MRI measures in order to have a comparison for the reliability of oscillation imaging. To that end, gas exchange images were analyzed following standard methods [13] to generate measures including RBC/Membrane, Membrane/Gas, RBC/Gas, and RBC Defect percent. These measures were compared across scan/rescan to assess their same-day reliability using intra-class correlation coefficients such that this reliability can be compared with oscillation imaging reliability.

## Oscillation imaging

**Binning algorithms.** The present work continues the development of binning methods for identifying projections to be used for keyhole image reconstruction. In the initial implementation, peaks and valleys of the RBC oscillation were normalized and a 20% threshold was used to select projections to be included in keyhole reconstruction [7]. However, this 20% threshold method is susceptible to incorrect binning of noise spikes, so improvements were needed. In the next iterations, a peak finding algorithm was used to identify peaks and valleys and a consistent number of projections per cycle was selected around each of these peaks and valleys [8,9]. However, the specific peak-finding algorithm was not deeply investigated especially as it pertains to accurately identifying peaks and valleys in the noisy RBC signal. We investigated this through the following peak finding methods.

The first point on each radial projection ($k_0$) of the dissolved xenon image was used for binning. Due to the non-equilibrium magnetization of hyperpolarized xenon, the signal at $k_0$ decays throughout the breath-hold. As such, the dissolved signal was first detrended using a double-exponential fit. After detrending, a phase shift was applied to separate RBC and membrane components. Subsequent binning steps were performed on the RBC $k_0$ trace.

In the first binning method tested, data were smoothed and fit to a multi-component sine fit (sum of 5 sine terms) prior to peak selection using a minimum peak separation of 0.6 times the period of the dominant oscillation, which should correspond to one cardiac cycle. Note that the dominant oscillation was determined by the location of the largest peak in the Fourier transform of the data. In the second binning method, data were bandpass-filtered using a window of 0.5–2 Hz [5], followed by peak selection using the same minimum peak separation. In both cases, algorithms were run both in a fully automated fashion and with the option for manual editing of the peak location. After peaks and valleys were selected, the global oscillation amplitude ($\alpha_{k0}$) was calculated following the equation:

$$\alpha_{k0} = \frac{\overline{k}_{0,maxima} - \overline{k}_{0,minima}}{\overline{k}_0}$$

In this equation, $\overline{k}_{0,maxima}$ is the mean RBC $k_0$ signal from oscillation maxima, $\overline{k}_{0,minima}$ is the mean RBC $k_0$ signal from oscillation minima, and $\overline{k}_0$ is the mean RBC $k_0$ signal.

The reliability of global oscillation amplitude for each binning method was assessed using intraclass correlation coefficients. Subsequent algorithm development was restricted to the binning method that provided the most reliable global oscillation amplitude measurement.

**Examination of the quality of oscillations.** Of the commonly measured Xe-MRI contrasts, the RBC signal has the lowest signal-to-noise ratio (SNR) [14]. Furthermore, most pulmonary diseases are associated with a reduction in RBC signal due to impaired gas exchange [15], further reducing the SNR of the RBC signal. Finally, Oscillation amplitudes are

commonly in the range of 5–20% of this already weak RBC signal [4]. As a result, RBC oscillations are often obscured by noise. This is particularly the case in PAH patients, where both the overall RBC signal and oscillation amplitude are reduced. To interrogate the impact of noise on oscillation imaging, a visual grading scale (Fig 1) was used to assess the quality of oscillations.

In addition, quantitative measures of oscillation quality were generated by performing a Fourier transform of the RBC signal trace, performing a Lorentzian fitting of the peak, and quantifying the frequency, SNR, and full width at half maximum (FWHM). Note that SNR was defined as the maximum intensity of the peak divided by the mean of the signal measured at the edge of the frequency range. The oscillation frequency was converted into a heartrate (HR), which was compared to the heartrate recorded by a monitor during imaging. Specifically, the absolute value of the difference between the recorded HR and the oscillation-based HR was used ("HR difference"). A composite score was calculated from FWHM, SNR, and HR difference. Briefly, each value was calculated as a z-score. Then, because quality is presumed to be optimal with minimized FWHM and heart rate difference and maximized SNR, the composite score ($Q_{comp}$) was calculated as:

$$Q_{Comp} = Z_{FWHM} + Z_{HR\ difference} - Z_{SNR}$$

Each of the three quality measures as well as the composite score were compared with visual grading, and they were incorporated into a regression model to determine whether oscillation quality metrics could be used to predict the reliability of oscillation amplitude.

**Oscillation imaging.** After peak binning, the selected projections were used for keyhole image reconstruction. Following the initial implementation, a two-key approach was used to generate images. Additionally, a more recently proposed sliding window keyhole reconstruction was used to generate a continuous set of keys.

Based on previously published simulations, a uniform keyhole radius of 9 points along the radial arm was used for all image sets [8]. For the two-key approach, images of regional oscillation amplitude were generated by taking the difference between the high key and low key on a per-voxel basis. For the multi-key approach, oscillation images were generated voxel-by-voxel using the difference between the key with the largest value and the key with the smallest value (for every voxel).

To provide a measure of amplitude that is comparable across different participants with differing RBC signal, this difference between high and low keys must be scaled. In the original implementation, the overall mean RBC signal was used for this scaling. Subsequent iterations to the two-key approach instead scaled by the RBC signal on a per-voxel basis. Both methods are explored in this work.

$$Amp_{bymean,x,y,z} = \frac{HighKey_{x,y,z} - LowKey_{x,y,z}}{Mean(RBC)}$$

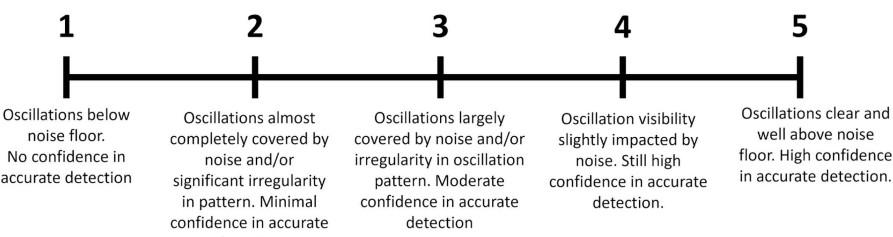

**Fig 1. Oscillation grading scale. Oscillations in the raw RBC signal were graded from 1 (no confidence in visibility of oscillation) to 5 (high confidence in visibility of oscillation).**

$$Amp_{byvoxel,x,y,z} = \frac{HighKey_{x,y,z} - LowKey_{x,y,z}}{RBC_{x,y,z}}$$

Finally, the most recent implementation using multiple keys to map both oscillation and phase used per-voxel scaling. Thus, we ultimately evaluated three distinct methods to generate oscillation maps: (1) 2 keys, scaled by mean of RBC signal ("2-key, mean-scaled"); (2) 2 keys, scaled voxel-wise by RBC signal ("2-key, voxel-scaled"); (3) Multiple keys ("Multi-key"). Different oscillation mapping methods were compared using a variety of techniques.

Oscillation amplitude maps using all three methods from the healthy volunteers were used to generate a healthy reference dataset. Variation within the lungs for each technique was calculated by generating the coefficient of variation within the lungs of each healthy volunteer. The first and second tertile of oscillation amplitude values for the healthy reference in each mapping method were calculated. These tertiles were used to bin oscillation maps from PAH and SSc patients into "high", "middle", and "low" values.

To examine reliability, whole-lung means of the different oscillation amplitude measures were compared between the first and second scans. To examine differences between scans on a regional basis, the second scan was registered to the first and structural similarity index (SSIM) and mean square error (MSE) were used. Due to the wider variability in amplitude values in the multi-key method, images were converted to modified z-scores prior to calculating the MSE. Modified z-scores were used because multi-key images had a strongly non-normal distribution that led to skewed z-scores. Briefly, the median of the healthy reference volunteers was calculated and used to calculate the median absolute deviation for the healthy reference values. Subsequently, every voxel in images was converted to a modified z-score based on the healthy reference values and median absolute deviation. In addition, oscillation maps were binned into high, middle, and low values as described above. Label masks corresponding to each of these bins (high, middle, low) were then compared between baseline and follow-up using the average distance metric. Average distance is a commonly used metric to compare segmentations [16], with a value of 0 indicating perfect agreement. Dice coefficients, though commonly used, were not employed in this work because they perform poorly on segmentations with few voxels, which was the case for some of the bins.

In addition to reliability analysis, means and standard deviations of oscillation amplitude metrics using the three mapping techniques were also compared across the three patient groups studied (healthy, PAH, SSc). Differences in phase mapping were compared for the multi-key method. Comparisons were made using 1-way ANOVA tests with post-hoc Tukey tests. Post-hoc testing was only performed if ANOVA testing returned a significant p-value ($p < 0.05$).

Finally, the multi-key method was compared to the 2-key, voxel-scaled method. Specifically, the multi-key method produces both oscillation amplitude and phase maps. In theory, scaling the oscillation amplitude by the cosine of the phase should reproduce the 2-key, voxel-scaled result. This calculation was performed, and the resulting images were compared using SSIM.

## Results

### Participants

Datasets were available for all 25 participants. Basic demographic information include: Healthy: N = 9, N = 5 Female, Age = 31.6 ± 4.1; SSc: N = 6, N = 4 Female, Age = 55.3 ± 14.0; PAH: N = 10, N = 8 Female, Age = 52.4 ± 11.6. All SSc participants had two image sets within the same imaging session. 9 out of 10 PAH patients had two datasets within the same imaging session. All (N = 10) PAH patients had data available for a second visit at 6-weeks to assess 6-week reliability in this population. However, 2 of these patients had significant medical events (1 hospitalization, 1 COVID-19 diagnosis) in the time between baseline and 6-week scans. These events may impact oscillation measures, so these 2 participants were omitted from 6-week reliability analysis.

## Reliability of standard gas exchange measures

The reliability of standard gas exchange measures is shown in Fig 2. These measures showed a high degree of scan-rescan reliability. RBC/Membrane had the highest reliability, with ICC = 0.98 (p < 0.001). Membrane/Gas had similar reliability, with ICC = 0.94 (p < 0.001). RBC measures had slightly greater scatter, but still showed good quantitative reliability (RBC/Gas: ICC = 0.91, p < 0.001; RBC Defect Percent: ICC = 0.83, p < 0.001). Reliability was higher for PAH participants as compared with SSc Participants (**PAH** – RBC/Membrane: ICC = 0.98, p < 0.001; Membrane/Gas: ICC = 0.99, p < 0.001; RBC/Gas: ICC = 0.96, p < 0.001; RBC Defect Percent: ICC = 0.93, p < 0.001; **SSc** - RBC/Membrane: ICC = 0.92, p < 0.001; Membrane/Gas: ICC = 0.75, p = 0.02; RBC/Gas: ICC = 0.77, p = 0.01; RBC Defect Percent: ICC = 0.54, p = 0.09).

## Impact of binning algorithm on global oscillation amplitude reliability

Differing binning methods showed drastic differences in the ability to resolve oscillations. While smoothing and sine-fitting helped to reveal oscillations that were obscured by noise, there were many examples of participants with no clear oscillations. Bandpass filtering, on the other hand, was able to clearly reveal oscillations, even in participants whose raw RBC data showed no discernable oscillation prior to processing. These qualitative observations were borne out in reliability analysis, where global oscillation amplitude showed far greater reliability for bandpass filtering methods (Fig 3). Using basic smoothing with a sine fit and no manual intervention had the worst reliability (ICC = 0.66; p = 0.003), though manual intervention improved this somewhat (ICC = 0.83; p < 0.001). Bandpass filtering, both for automatic (ICC = 0.87; p < 0.001) and manual intervention (ICC = 0.88; p < 0.001) cases showed excellent reliability. Based on this, binning was performed using bandpass filtering with manual intervention for all analyses moving forward.

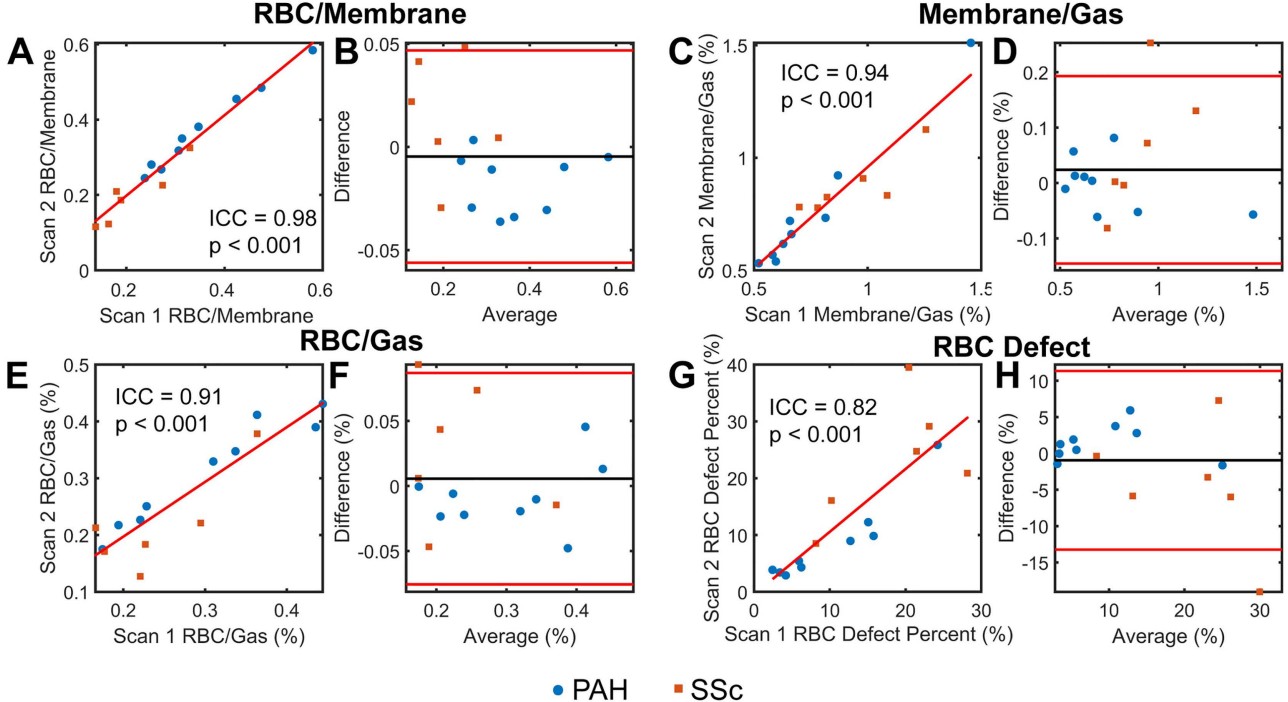

**Fig 2. Reliability of standard Xe-MRI quantitative metrics.** RBC/Membrane (A) shows the strongest reliability (ICC = 0.98), though all measures, including Membrane/Gas (C), RBC/Gas (E), and RBC Defect Percent (G) have ICC values greater than or equal to 0.82, implying strong reliability of these measures. Bland Altman plots (B, D, F, H) are shown alongside correlation plots to emphasize the good agreement between measures.

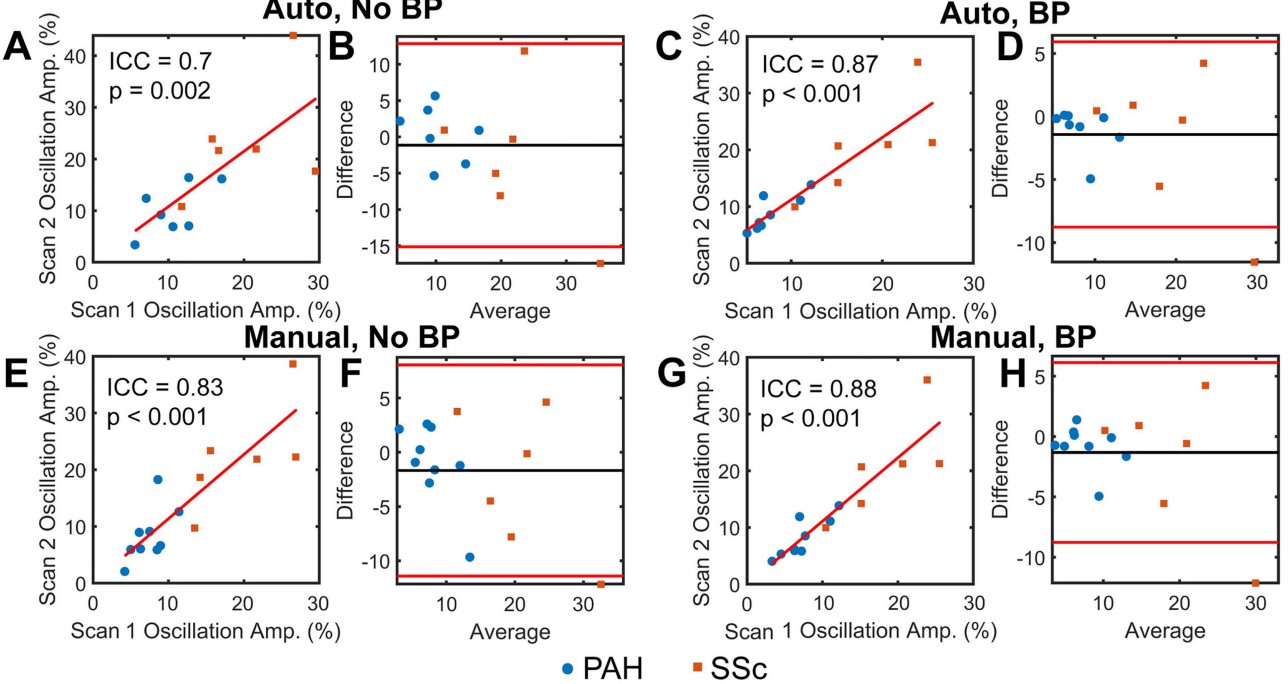

**Fig 3. Reliability of Global RBC Oscillation Amplitude for different binning strategies.** When a bandpass filter is not used (A, B, E, F), it is more challenging to accurately identify oscillation peaks and valleys, which leads to worse reliability, even when manual intervention is implemented (ICC = 0.7 for automatic detection, ICC = 0.83 for manual intervention) (E). When a bandpass filter is used, oscillation peaks and valleys are able to be detected, and reliability is strong for both automated (C, D; ICC = 0.87) and manual intervention cases (G, H; ICC = 0.88). Bland Altman plots (B, D, F, H) are shown alongside correlation plots to highlight improved agreement between scans for bandpass filtering methods. Abbreviations: BP – Bandpass Filter.

### Impact of oscillation quality on oscillation amplitude reliability

To the naked eye, RBC signal oscillations varied widely in quality, with the lowest quality oscillations occurring in PAH patients (mean reader score of 2.6) and the highest quality in healthy volunteers (mean reader score 4.0) and SSc patients (mean reader score 3.9). Examples of different oscillation traces and their grades are shown in Fig 4. Additional quality metrics were calculated including the FWHM and SNR of the dominant oscillation frequency peak, and the difference between the frequency of oscillations and the participant's measured heart rate. Generally, these quantitative measures of oscillation quality agreed with reader grading (Fig 5): FWHM was largest for poor quality scores; Heart rate difference was largest for poor quality scores; SNR was largest for high quality scores. A composite score based on these three quantitative measures was able to successfully differentiate between reader scores.

When considering only univariate analysis, the correlation between scan 1 and scan 2 global oscillation amplitude was strong (R = 0.907, p < 0.001). The addition of individual oscillation quality metrics as covariates improved the quality of the regression model (Reader score: R = 0.918; FWHM: R = 0.909; HR difference: R = 0.957; SNR: R = 0.972; Composite Score: R = 0.962; all p < 0.001). When examining specific covariates, only HR difference was a significant predictor in the model (p = 0.049), though interaction terms for SNR (p = 0.006) and composite score (p = 0.01) were significant within the models. Similar findings were observed for whole-lung mean values of oscillation maps, with the inclusion of quality measures generally improving the correlation coefficient but not emerging as significant predictors within the models.

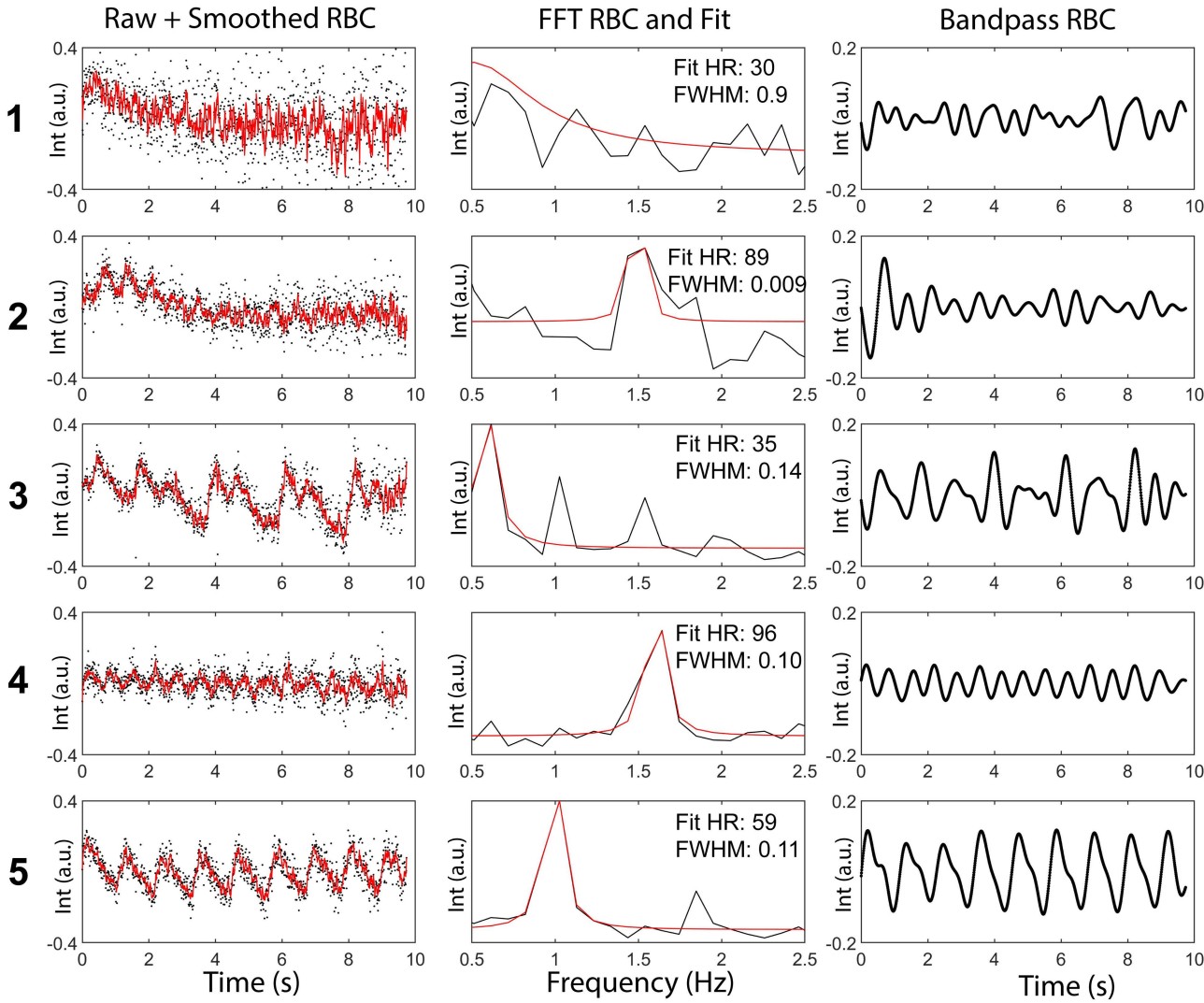

**Fig 4. Representative RBC oscillation traces for each of the grades 1–5, 1 being the worst quality and 5 being the highest.** Column 1 shows the raw RBC signal and the smoothed oscillation trace. Column 2 shows the Fourier transform of the RBC oscillation, with heart rate (HR) and full width at half maximum (FWHM) provided. Column 3 shows bandpass-filtered RBC oscillations, demonstrating a clear improvement in the ability to resolve oscillation peaks and valleys.

## Oscillation imaging

Example images are shown in Fig 6. Different algorithms lead to significantly different image characteristics, different variability, and different ranges of values. The mean-scaled method had the least variation across the lungs, with a mean coefficient of variation of 0.51 in healthy volunteers. Voxel-scaled oscillation maps had intermediate variation, with a mean coefficient of variation of 0.71 in healthy volunteers. The multi-key method had the greatest variation, with a coefficient of variation of 1.29 in healthy volunteers. 2-key methods lead to a range of values very similar to those observed in global oscillation measurements (~0–20%). In contrast, the multi-key method leads to much larger values (~0–100%). However, when the multi-key amplitude map is scaled by the cosine of the phase map, the values of the resulting map are brought down to a similar range to those observed in the two-key method. In addition, the features of these scaled maps are

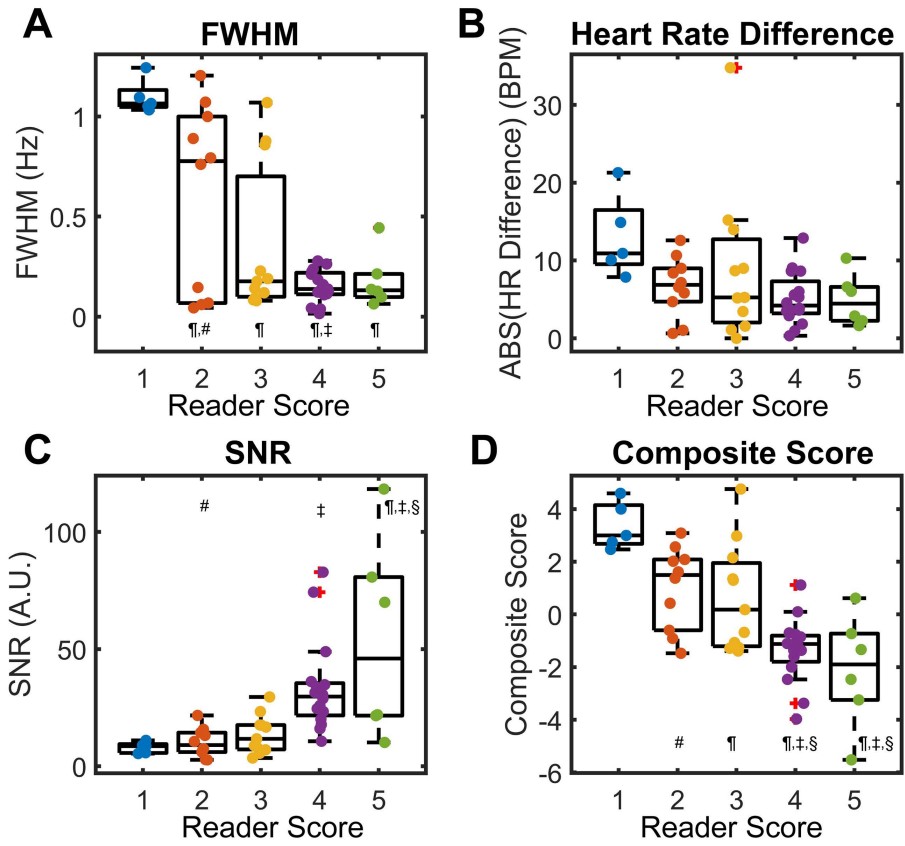

**Fig 5. Comparison of quantitative quality metrics to reader scoring.** (A) Full width at half maximum (FWHM) of the dominant oscillation peak was largest for low scores. (B) Heart rate (HR) difference was similarly largest for low scores. (C) Signal-to-noise ratio was largest for high scores. (D) A composite score composed of these three measures effectively discriminated between the poorest oscillation quality (Score of 1), moderate oscillation quality (Score of 2 or 3), and good oscillation quality (Score of 4 or 5). Symbols signify significant differences ($p < 0.05$): ¶: versus 1; ‡: versus 2; §: versus 3; #: versus 4.

generally similar to those of the 2-key, voxel-scaled maps (Fig 7). Notably, images scaled voxel-wise by the RBC signal (2-key, voxel-scaled and multi-key) exhibit a gradient from anterior to posterior, with amplitude higher in the anterior and lower in the posterior. Maps generated by scaling by the mean RBC signal (2-key, mean-scaled), exhibit an opposite anterior-to-posterior gradient, with amplitudes higher in the posterior and lower in the anterior (Fig 8). Apical to basal gradients are more subtle and the direction of gradient is the same across all three methods.

The different oscillation imaging methods each showed ability to discriminate between different conditions. When examining global oscillation amplitude, SSc participants had significantly higher oscillation amplitude than healthy ($p < 0.001$) and PAH participants ($p < 0.001$) (Fig 9). PAH patients generally had lower global oscillation amplitude than healthy volunteers, but this failed to reach statistical significance ($p = 0.09$). Both 2-key methods showed very similar results, with SSc participants showing larger oscillations and PAH participants showing lower oscillations. These features are still present, but somewhat suppressed for the multi-key method. This is particularly the case for PAH participants, whose mean oscillation amplitude for the multi-key method largely overlap with values observed in healthy volunteers. Differences between healthy and PAH participants are largely recovered in the multi-key method if the amplitude is scaled by the cosine of the phase (Fig 9).SSc participants exhibited larger standard deviation of oscillation values than healthy volunteers ($p = (0.03)$) and PAH participants ($p = 0.02$) for the 2-key, mean scaled method, and greater standard deviation than PAH participants

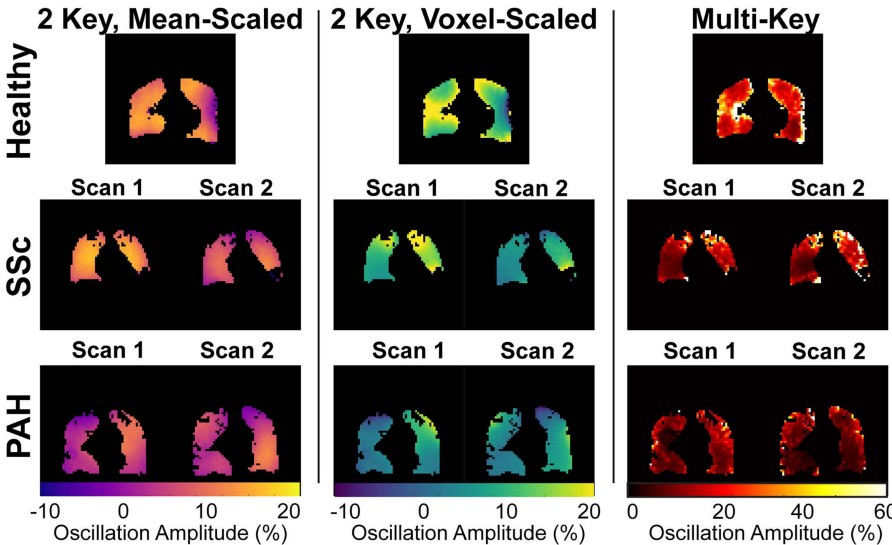

**Fig 6. Central coronal slices for representative oscillation amplitude maps for the 3 algorithms and 3 participant groups investigated in this study.** For SSc and PAH patients, the two scans that occurred back-to-back on the same day are shown. In the healthy participants, both 2-key methods provide similar results, but regional differences are observed in SSc and PAH patients, likely due to regional variation in RBC signal. Multi-key images show significantly different features and much larger values for oscillation amplitude, owing to the phase information also obtained by that method.

for the 2-key, voxel scaled (p = 0.02) and multi-key (p = 0.04) methods. Values extracted from phase maps alone (mean, median, standard deviation, interquartile range) were not significantly different across groups.

### Same-day oscillation imaging reliability

Whole-lung mean values for each imaging method showed relatively strong scan-rescan reliability when including data from both SSc and PAH participants: 2-key, mean-scaled: ICC = 0.87, p < 0.001; 2-key, voxel-scaled: ICC = 0.88, p < 0.001; multi-key: ICC = 0.86, p < 0.001. Similarly, the percentage of images binned into high, low, and mid tertiles based on a healthy reference also exhibited relatively high ICC (See Table 1). When examining participants by disease, reliability of whole-lung mean and bin percents was stronger in patients with PAH than in those with SSc (Table 1).

Image comparison metrics were also implemented to better capture the regional information from oscillation amplitude maps. Scan-rescan maps had relatively high SSIM for each of the three methods, with the highest values observed for the 2-key, mean-scaled method (mean SSIM = 0.90) and lowest values for multi-key (mean SSIM = 0.87) (Fig 10). MSE showed similar results, with the lowest MSE for the 2-key, mean-scaled method (mean MSE = 0.23) and the highest MSE for the multi-key method (mean MSE = 1.19). Comparing the high, mid, and low-binned maps using the average distance metric showed reasonable agreement for each of the methods. Average distance was consistently lowest for the multi-key method and highest for the 2-key, voxel-scaled method.

When examining these regional comparison metrics in the context of oscillation quality metrics, there were no correlations between SSIM or MSE and oscillation quality metrics. When examining average distance for high binned oscillations, there was a significant correlation between average distance and reader score for all three mapping methods (Mean-scaled: R = −0.85; Voxel-scaled: R = −0.62; Multi-key: R = −0.62). Average distance for high-binned oscillations and the composite quality score also correlated significantly for mean-scaled (R = 0.82) and multi-key (R = 0.61) methods. Average distance for low- and middle-binned oscillations did not correlate with any oscillation quality metrics.

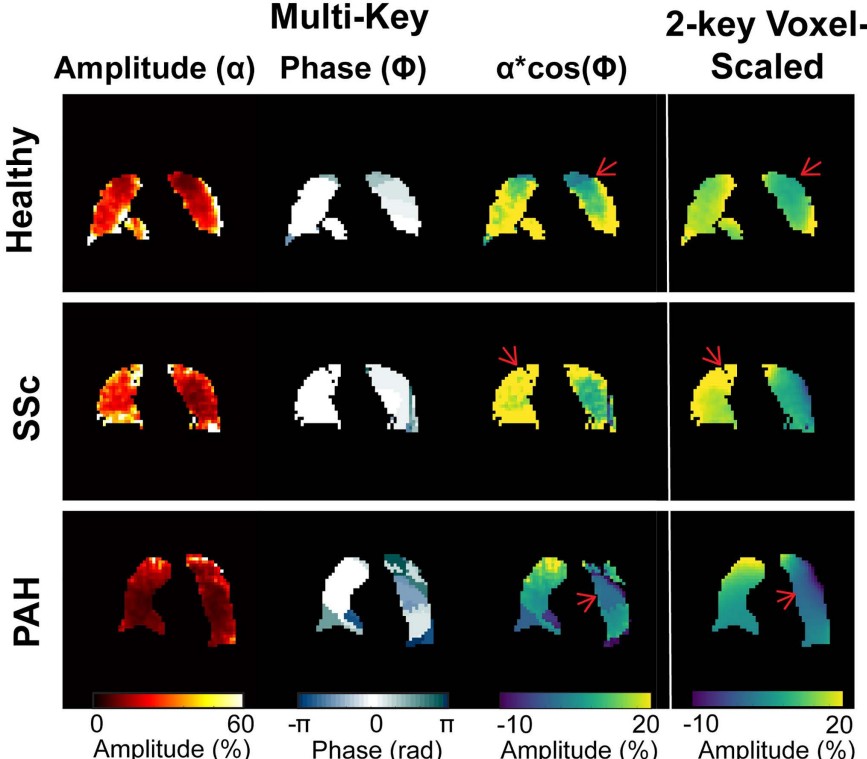

**Fig 7. Central coronal slice from representative participants demonstrating that 2-key voxel-scaled amplitude maps can be largely reproduced using the multi-key data.** Scaling the amplitude by the cosine of the oscillation phase brings the range of values to the same as the 2-key method and largely reproduces regional differences in oscillation amplitude.

For the multi-key method, it is also possible to examine the reliability of phase maps. For all phase maps generated, both the mean and median phase are very close to zero. As such, the calculation of ICC is drastically impacted by small deviations and thus ICC returns a value close to 0 (i.e., no correlation). Instead, SSIM and MSE have greater value in this context. SSIM between phase-maps on the same day averaged $0.79 \pm 0.23$ while MSE averaged $0.78 \pm 1.8$ radians.

### Six-week oscillation imaging reliability

In PAH patients, reliability of oscillation imaging at 6-weeks follow-up was assessed. Global oscillation amplitude reliability was high (ICC = 0.92, p < 0.001). However, regional oscillation amplitude showed lesser reliability, with each of the methods showing moderate to low ICC for whole-lung means and bin percentages (Table 1). Despite this, SSIM values for the two scans six weeks apart were relatively high (2-key, mean-scaled: mean SSIM 0.89; 2-key, voxel-scaled: mean SSIM 0.87; multi-key: mean SSIM 0.87). MSE for 6-week comparisons was similar to those observed for same day: MSE for the 2-key, mean-scaled method was 0.72, for the 2-key, voxel-scaled was 0.77, and for the multi-key method was 2.12. Average distance comparing bin maps was also similar to same-day measures, with average distance consistently lowest for the multi-key method and highest for the 2-key, voxel-scaled method.

### Discussion

RBC oscillation amplitude mapping using keyhole image reconstruction is a technique still in its infancy. There have been 3 publications using this technique [7–9], with each implementing differing algorithms for the generation of these maps.

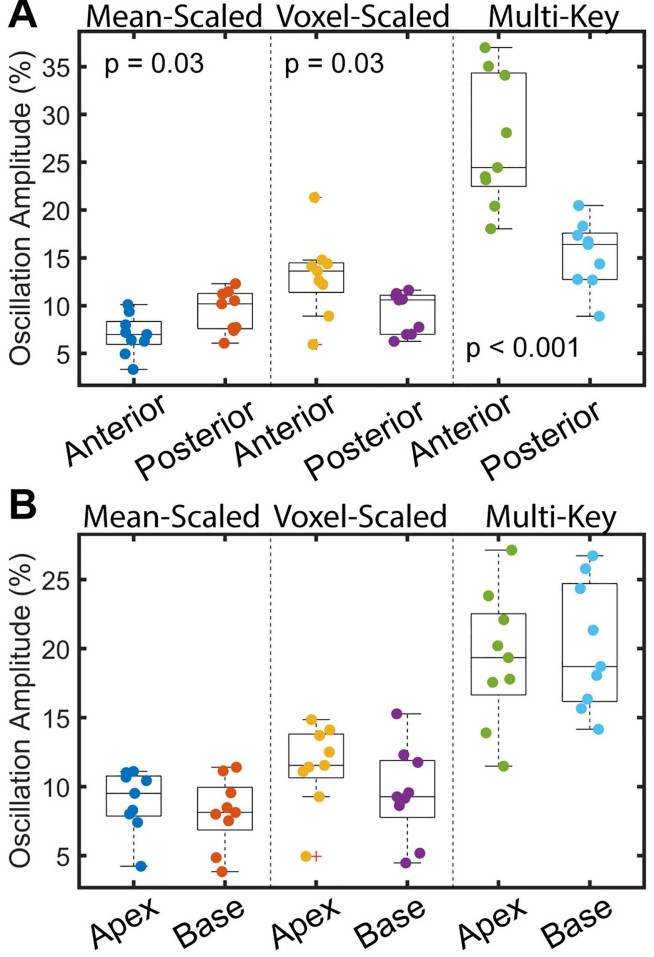

**Fig 8. Boxplots showing differences in oscillation amplitude in the anterior vs. posterior (A) and in the apex vs base (B) of the lungs for healthy volunteers.** There is a clear difference in oscillation amplitude between the anterior and posterior of the lungs, though the direction of this difference is opposite for the mean-scaled method as compared to the voxel-scaled and multi-key methods. There is no significant difference in oscillation amplitude between the apex and base of the lungs.

In this work, we have comprehensively evaluated these different techniques in young, healthy volunteers as well as SSc and PAH patients. These three groups provided data that represented "ideal" oscillation amplitude (healthy volunteers), high oscillation amplitude (SSc), and low oscillation amplitude (PAH). Investigating this wide range of physiology and pathophysiology led to a broad variation in oscillation quality and was essential for testing the different oscillation mapping methods.

PAH patients consistently had weak oscillations that were often difficult to visualize over noise, while SSc patients and healthy volunteers had much clearer oscillations. Reader scoring of oscillations as well as quantitative measures of oscillation quality were implemented to assess whether "oscillation quality" would be predictive of oscillation reliability. It appears that quantitative measures of oscillation quality, such as SNR, HR difference, and FWHM, can largely reproduce reader scoring, which has the potential to reduce interrater variability in quality control. However, our analyses suggest that, surprisingly, the quality of oscillations has a minimal impact on the reliability of oscillation mapping. The reliability of global and whole lung mean oscillation amplitude means did not depend significantly on oscillation quality measures.

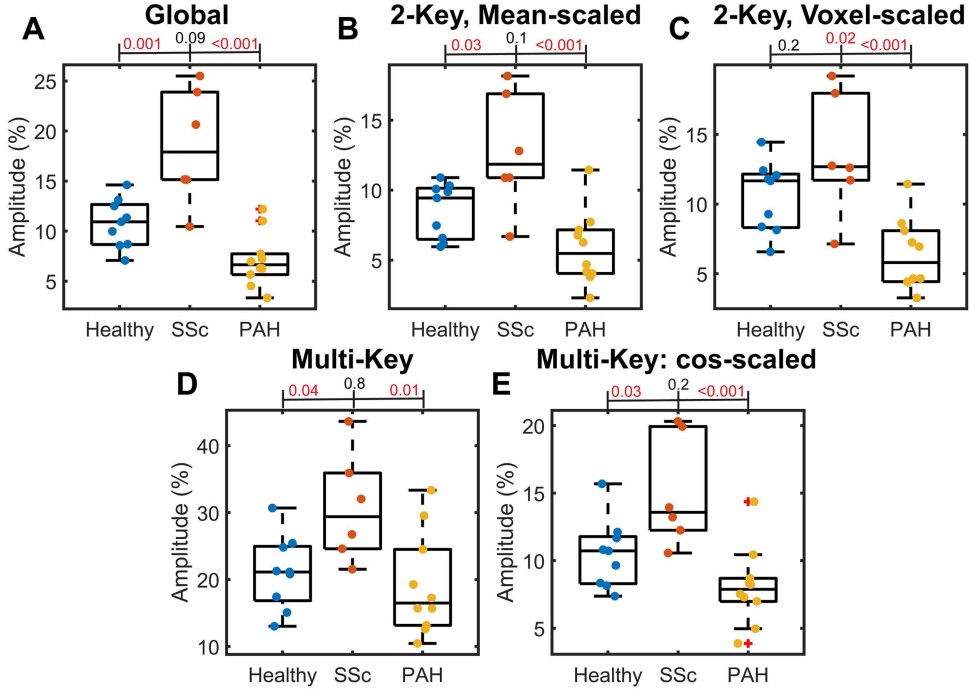

**Fig 9. Boxplots showing the difference in mean oscillation amplitude between healthy, PAH, and SSc participants for the different oscillation mapping techniques investigated: (A) Global, (B) 2-key, mean-scaled; (C) 2-key, voxel-scaled, (D) Multi-key, (E) Multi-key scaling mapped amplitude by the cosine of the mapped phase.** P-values from post-hoc testing are shown above plots, with significant (p < 0.05) values shown in red. The global measure of oscillation amplitude (A) provides the best discrimination between participant groups. Each of the oscillation mapping techniques leads to greater overlap, but, in all cases, the PAH patients exhibit the lowest amplitude and SSc the highest. Scaling multi-key amplitude by the cosine of the phase strengthens the separation between participant groups (E) as compared to just the multi-key amplitude.

Moreover, there were few correlations between regional measures (SSIM, MSE, and average distance) and oscillation quality measures. Specifically, only average distance measures comparing high-binned oscillation maps correlated with quality measures. While these correlations corresponded with expectation (improved reader score and improved composite score led to reduced average distance), it is unexpected that they are only present for high-binned data. That reliability of oscillation mapping appears not to be strongly affected by oscillation quality is surprising because the oscillation mapping technique depends on the accurate identification of peaks and valleys of the oscillation. As oscillation quality goes down, it would follow that the accuracy of detection of peaks and valleys would suffer, particularly in PAH patients in whom oscillation amplitudes are weak. Yet, this was not borne out in our results, with PAH patients showing greater reliability in oscillation than SSc patients, a counterintuitive result.

To isolate oscillations, previous work has used multiple different smoothing algorithms as well as bandpass filtering. In this work, we investigated a combined smoothing and fitting algorithm as well as bandpass filtering. Bandpass filtering was unequivocally the superior method: Bandpass filtering suppressed noise and made oscillations visible, even in the noisiest datasets. This success was clear in reliability analysis; oscillation amplitude calculated based on bandpass filtered data was far more reliable than that calculated from smoothed data. Notably, when using bandpass-filtered data, the reliability of global oscillation amplitude (ICC = 0.88) was comparable to the reliability of the RBC/Gas ratio (ICC = 0.91), a standard gas exchange Xe-MRI measure.

Each of the three different methods of oscillation mapping that were investigated were successfully used to generate amplitude maps for each participant. The 2-key methods both provided regional amplitude values in a similar range, while

**Table 1. Same-day ICC and p values for whole-lung mean oscillation amplitude as well as the percentage of the lungs binned to low, mid, and high values. Each comparison is done for all participants as well as for SSc and PAH participants individually. For PAH patients, the 6-week reliability data is also provided.**

| Comparison | Osc Mapping Method | Condition | | ICC | p-value |
|---|---|---|---|---|---|
| Scan-rescan | 2-key, mean-scaled | SSc and PAH | Mean | 0.87 | <0.001 |
| | | | Low | 0.80 | <0.001 |
| | | | Mid | 0.45 | 0.036 |
| | | | High | 0.88 | <0.001 |
| Scan-rescan | 2-key, voxel-scaled | SSc and PAH | Mean | 0.87 | <0.001 |
| | | | Low | 0.87 | <0.001 |
| | | | Mid | 0.29 | 0.11 |
| | | | High | 0.84 | <0.001 |
| Scan-rescan | Multi-key | SSc and PAH | Mean | 0.87 | <0.001 |
| | | | Low | 0.88 | <0.001 |
| | | | Mid | 0.91 | <0.001 |
| | | | High | 0.87 | <0.001 |
| Scan-rescan | 2-key, mean-scaled | SSc | Mean | 0.78 | 0.013 |
| | | | Low | 0.53 | 0.093 |
| | | | Mid | 0.28 | 0.25 |
| | | | High | 0.68 | 0.033 |
| Scan-rescan | 2-key, voxel-scaled | SSc | Mean | 0.79 | 0.011 |
| | | | Low | 0.74 | 0.019 |
| | | | Mid | −0.085 | 0.56 |
| | | | High | 0.62 | 0.054 |
| Scan-rescan | Multi-key | SSc | Mean | 0.61 | 0.057 |
| | | | Low | 0.11 | 0.4 |
| | | | Mid | 0.78 | 0.01 |
| | | | High | 0.60 | 0.06 |
| Scan-rescan | 2-key, mean-scaled | PAH | Mean | 0.76 | 0.004 |
| | | | Low | 0.71 | 0.008 |
| | | | Mid | 0.53 | 0.047 |
| | | | High | 0.87 | <0.001 |
| Scan-rescan | 2-key, voxel-scaled | PAH | Mean | 0.80 | 0.002 |
| | | | Low | 0.80 | 0.002 |
| | | | Mid | 0.67 | 0.01 |
| | | | High | 0.85 | <0.001 |
| Scan-rescan | Multi-key | PAH | Mean | 0.85 | <0.001 |
| | | | Low | 0.88 | <0.001 |
| | | | Mid | 0.91 | <0.001 |
| | | | High | 0.80 | 0.002 |
| Baseline-6-weeks | 2-key, mean-scaled | PAH | Mean | 0.76 | 0.006 |
| | | | Low | 0.48 | 0.08 |
| | | | Mid | 0.35 | 0.16 |
| | | | High | 0.74 | 0.008 |
| Baseline-6-weeks | 2-key, voxel-scaled | PAH | Mean | 0.81 | 0.003 |
| | | | Low | 0.69 | 0.02 |
| | | | Mid | 0.21 | 0.28 |
| | | | High | 0.88 | <0.001 |

*(Continued)*

**Table 1.** (Continued)

| Comparison | Osc Mapping Method | Condition | | | ICC | p-value |
|---|---|---|---|---|---|---|
| Baseline-6-weeks | Multi-key | PAH | | Mean | 0.26 | 0.23 |
| | | | | Low | 0.57 | 0.04 |
| | | | | Mid | 0.60 | 0.04 |
| | | | | High | 0.43 | 0.11 |

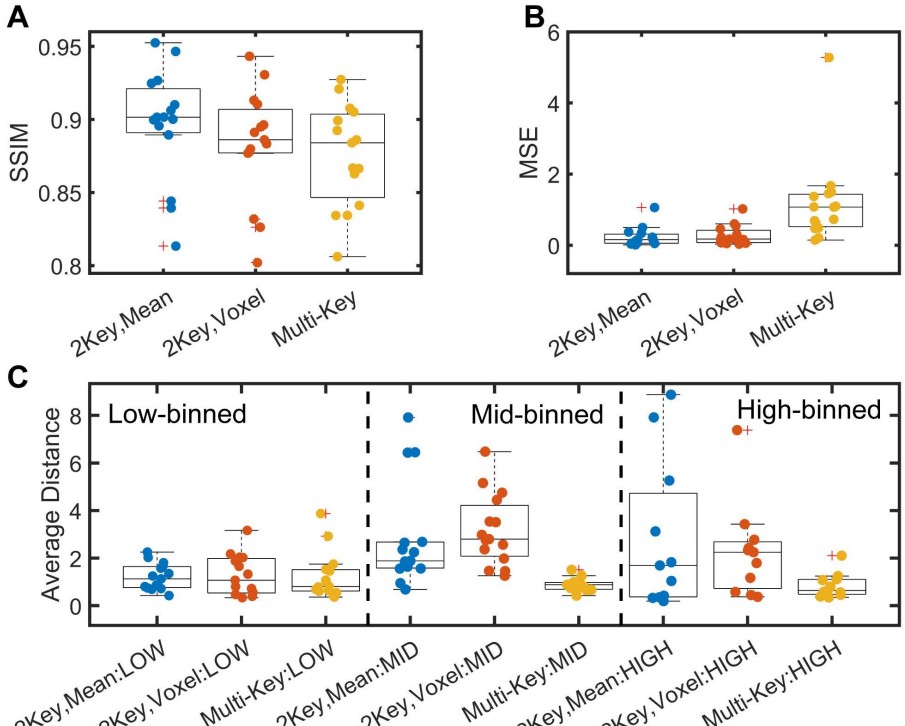

**Fig 10. Comparison of regional image comparison metrics for two back-to-back images for the different oscillation mapping techniques investigated.** (A) Structural Similarity index is generally high, with the best values observed for the 2-key, mean-scaled maps. (B) Mean square error is relatively low for the 2-key methods but very large for the multi-key method. This appears to be driven by a small number of significant outliers. (C) The average distance metric shows that there is reasonable overlap between regions binned to low, mid, and high values.

the multi-key method gave much larger regional amplitude values. As discussed in the original presentation of the multi-key method [9], these differences are due to phase differences being incorporated into the 2-key amplitude measurement, while phase is separately mapped in the multi-key method. When the amplitude and phase maps from the multi-key method are combined, amplitude maps very similar to the 2-key, voxel-scaled method are generated.

Each of the oscillation mapping techniques showed differences across the three conditions studied (healthy, SSc, and PAH). As expected, PAH patients exhibited reduced oscillations, consistent with upstream pre-capillary impedance to blood flow. SSc patients exhibited elevated oscillations, which is consistent with values observed in idiopathic pulmonary fibrosis [7]. Recent work suggests that this elevation in oscillation amplitude may be due to reduced RBC transfer with preserved capillary blood volume oscillations [10], which would be consistent with the low RBC transfer we observed in SSc participants. The 2-key methods most clearly differentiated between these conditions, while amplitude alone from the

multi-key method showed lesser differences across groups. This finding agrees with previous work [9] and suggests that differences in global and 2-key oscillation amplitude measurements include contributions both from blood volume oscillations within the capillary bed and from regional changes to the timing of the cardiac pulse wave.

Another relevant finding in comparing the different oscillation mapping techniques is that regional variation in oscillation amplitude is different depending on whether the amplitude is scaled by the whole-lung mean RBC signal or by the RBC signal in each individual voxel. Most notably, anterior-to-posterior gradients are reversed between the voxel-scaled and mean-scaled cases. This is due to the known gravitational gradients observed in gas exchange Xe-MRI, where the RBC signal is lower in the anterior of the lungs when participants are imaged in the supine position. With lesser RBC signal in the anterior, scaling by the whole lung mean leads to non-physiologically small values of oscillation amplitude. Scaling voxel-wise corrects for this regional variation in RBC signal and appears to provide a more physiologically accurate measurement of regional RBC oscillation amplitude, though at the expense of slightly more variation in values across the lungs.

Originally, scaling by the mean of the overall RBC signal was proposed to avoid causing unreasonably large oscillation values by scaling by very low RBC signal. This is particularly a concern in patients with pulmonary disease, who often exhibit decreased RBC signal. However, improvements in scanning techniques and polarization levels are mitigating these concerns. Moreover, calculating oscillation amplitude only in voxels that have RBC signal above the defect level also protects against this. Evidence that this problem is mitigated is found in the ICC values observed for the 3 techniques in the scan-rescan comparison. For whole-lung mean oscillation amplitude, all three techniques had ICC in the same range (0.86–0.88), with the 2-key, voxel-scaled showing the highest ICC. Likewise, SSIM for scan-rescan comparisons was relatively high for all three types of imaging. MSE was relatively low for all three methods, with the 2-key methods having MSE near 0.3 and the multi-key method having MSE near 1.2 (in units of modified z-score). It should be noted that the 2-key methods return amplitude maps with a roughly normal distribution, while the multi-key method returns a highly skewed distribution. As such, the higher MSE for the multi-key is likely due to differences in values on the high end of this non-normal distribution.

When moving beyond same-day scanning, the reliability of oscillation mapping seemed to suffer. While global oscillation amplitude showed relatively strong reliability (ICC = 0.89), whole lung means of oscillation maps were less robust (ICC = 0.81 for 2-key methods and ICC = 0.26 for multi-key). Further investigation is necessary to understand and mitigate the source of this poor long-term reliability.

Ultimately, the high scan-rescan reliability of these different oscillation mapping techniques strongly suggests that Xe-MRI oscillation imaging is detecting real physiology. However, the comparatively poorer reliability at 6-weeks in stable PAH patients indicates that continued development is necessary to implement these analyses more broadly.

While reliability is an important consideration in determining which of these oscillation mapping methods to use, there are other factors that should be noted. For example, 2-key methods return negative values, which do not have physiological meaning. As discussed by Pilgrim-Morris et al. [9] these non-physiological negative values likely reflect differing phase of cardiac pulse wave. Thus, the multi-key method that generates both amplitude and phase maps may have an advantage over 2-key methods. Other key considerations may include factors such as: computational efficiency (2-key methods have an advantage); Sensitivity to regional disease (some evidence for 2-key methods in Chronic Thromboembolic Pulmonary Hypertension, but more work using both methods is needed) [8]; and detection of disease progression/therapeutic change (little data for either).

This study has a number of limitations. Our numbers were relatively small, particularly within individual participant groups. Moreover, our healthy volunteers were not age-matched to patients with SSc or PAH. This is only a minor limitation, as the main goal of this work was to investigate the different oscillation mapping techniques as opposed to using the technique to differentiate between conditions. A larger limitation is that we were unable to collect 2 scans in healthy volunteers, so reliability analysis is limited to patients with pulmonary disease.

## Conclusions

RBC oscillation amplitude mapping is the newest contrast available to Xe-MRI researchers, but it has seen relatively little development. We have investigated three different oscillation mapping methods, using reliability to determine the most effective techniques. Our results suggest that all three methods (2-key, mean-scaled; 2-key, voxel-scaled; multi-key) have relatively strong scan-rescan reliability, but much weaker 6-week reliability. We have further noted discrepancies in the regional features of the 2-key, mean-scaled method, and thus recommend that images be scaled on a per-voxel basis. The use of 2 keys to generate oscillation amplitude maps limits the ability to differentiate between amplitude vs. phase abnormalities, but it does appear to better discriminate between healthy individuals and those with lung disease. Using multiple keys enables the generation of amplitude and phase maps, which may provide a more in-depth view of pulmonary hemodynamics. Moreover, combining amplitude and phase information from multiple keys appears to recover sensitivity to disease. Ultimately, RBC oscillation amplitude mapping provides insight into pulmonary hemodynamics that is not achievable from any other non-invasive technique, and the findings of this work indicate that it can be a robust and reliable measure, even in patients with lung disease.

## Acknowledgments

We thank Dr. Jim Wild and Dr. Jemima Pilgrim-Morris for sharing computer code for the generation of multi-key amplitude and phase maps.

## Author contributions

**Conceptualization:** Peter J. Niedbalski.

**Data curation:** Ivina Mali, Peter J. Niedbalski.

**Formal analysis:** Ivina Mali, Bradie Frizzell, Steven Haworth, Peter J. Niedbalski.

**Funding acquisition:** Peter J. Niedbalski.

**Software:** Peter J. Niedbalski.

**Supervision:** Peter J. Niedbalski.

**Visualization:** Peter J. Niedbalski.

**Writing – original draft:** Peter J. Niedbalski.

**Writing – review & editing:** Ivina Mali, Steven Haworth, Peter J. Niedbalski.

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
