## [Decision Letter · Decision Letter 0]

12 Jul 2025

Dear Dr. Niedbalski,

Thank you for submitting your manuscript to PLOS ONE. After careful consideration, we feel that it has merit but does not fully meet PLOS ONE’s publication criteria as it currently stands. Therefore, we invite you to submit a revised version of the manuscript that addresses the points raised during the review process.

**ACADEMIC EDITOR: ** some small clarifications and additional discussion/stats/info are reqired. Limitations and clinical perspectives should be better emphasized. All issues raised by expert Reviewers are required. 

We look forward to receiving your revised manuscript.

Kind regards,

Vincenzo Lionetti, M.D., PhD

Academic Editor

PLOS ONE

Journal Requirements:

2. Please expand the acronym “NIH” (as indicated in your financial disclosure) so that it states the name of your funders in full.

3. Thank you for stating the following in the Competing Interests/Financial Disclosure section:

I have read the journal's policy and the authors of this manuscript have the following competing interests: Peter Niedbalski is a consultant for Polarean Imaging, Plc.

We note that one or more of the authors are employed by a commercial company: Polarean Imaging, Plc

Reviewers' comments:

Reviewer's Responses to Questions

**Comments to the Author**

1. Is the manuscript technically sound, and do the data support the conclusions?

Reviewer #1: Yes

Reviewer #2: Yes

2. Has the statistical analysis been performed appropriately and rigorously?

Reviewer #1: Yes

Reviewer #2: Yes

3. Have the authors made all data underlying the findings in their manuscript fully available?

Reviewer #1: Yes

Reviewer #2: Yes

4. Is the manuscript presented in an intelligible fashion and written in standard English?

Reviewer #1: Yes

Reviewer #2: Yes

Reviewer #1: The authors describe an assessment of repeatability/reliability for RBC oscillation imaging acquired as part of the gas exchange Xe-MRI acquisition. They acquired data in specific populations which are expected to represent the majority of the signal variations observed in patients. They also compare multiple RBC oscillation analysis methods to provide insight into which may be better suited for applications to disease with the goal to optimize the methods. Overall, the manuscript is well done but could use some minor improvements to fully support their findings.

Comments:

1. Line 118: I suppose this TR is the TR between the same compartments and not between RF pulses. Please clarify.

2. Line 139: The 20% threshold method does not appear to be assessed/considered here. Has this method already been shown inferior? If so, how/where? If not, consider including it for reference.

3. Line 147: To detrend the dissolved signal, have the authors considered normalizing by the gas? I believe this would be more robust as they will be significantly correlated. It won’t contain the downstream magnetization decay, but would those projections not be discarded anyway?

4. Line 150: What is mean by a multi-component sine fit? Sum of two sine waves?

5. Line 151: How was one cardiac cycle determined? From the measured heart rate or the data?

6. Line 157: Have the authors considered calculating by median rather than mean? It would be more robust to outliers, but I am not sure if that’s an issue.

7. Line 201: I was initially confused by the underscore v in the equations before realizing it was for voxel. Would it make more sense to not have it since they all are 3D maps? Or use (x,y,z) instead?

8. Line 218: What is the reason behind using only distance and not also the even more common Dice coefficient? I assume its due to Dices poor performance with small/thin structures, but it would be useful to know.

9. Line 233: With the reduced 6-week reliability performance, was it confirmed that none of the patients had a change in treatment/medication or other reasons for varying hypertension.

10. Line 241: Can you provide insight into why reliability was higher for PAH? That goes against my intuition of lower signal and lower oscillations being more prone to error.

11. Line 255: The success of the bandpass over the lowpass suggests a low frequency component being present. Would comment 3 help with this?

12. Line 277: How was SNR of a single frequency measured?

13. Line 300: In the model, HR difference was the true difference and not the abs() like mentioned earlier, correct?

14. Line 301: Could you clarify how HR difference was significant despite a p=0.49?

15. Line 331: Weren’t these supposed to be anova’s? If so, were they significant before assessing the post hocs?

16. Line 333:P=0.09 for both PAH vs healthy and PAH vs SSc?

17. Line 382: Why was mse and distance not assessed for the 6-week tests like the same day?

18. Line 399: I may have missed it, but I don’t see anywhere where the effect of data quality on each method was performed.

19. Line 432: I would like to see the values and stats for the gradients since they seem important.

20. Line 437: Did voxel wise reduce the variation of the signal within the lung or did it just make them appear more physiologically accurate?

21. Line 473: The multi-key approach with phase-based scaling seems to do just as well so is this true?

22. Fig 7: This may be in the referenced paper, but is there any insight into why the phase maps are not continuous? Is that an artifact of the method or is it expected to be physiological? I could see it having sharp transitions for regional “up-stream” issues but it all seems to be very digitized in the phase dimension.

23. Fig 9: I assume these metrics are all for the raw multi-key oscillations? I think it would be useful to clarify and/or include the scaled version. Also, if it’s the raw values, you should likely use a normalized version to not penalize the multi-key method which produces higher oscillation percentages.

Reviewer #2: Summary:

Periodic oscillations in the signal from 129Xe dissolved in the pulmonary capillary red blood cells (RBCs) originating from changes in capillary blood volume over the cardiac cycle have recently emerged as a potential biomarker of cardiopulmonary disease. The amplitude of the oscillations can be mapped regionally from 3D radial dissolved-phase 129Xe imaging data using a post-acquisition keyhole reconstruction. This manuscript aimed to optimise the oscillation mapping method. The authors present a comparison of three published variations on the keyhole mapping method and evaluate their same-day and 6-week repeatability. Each method was able to detect differences in the oscillation amplitude of patients with pulmonary arterial hypertension and systemic sclerosis. This is an important contribution to the limited literature on RBC oscillation mapping and has several strengths, including:

• Implementation of the oscillation mapping technique to two patient groups with reduced and enhanced oscillation amplitude respectively, including first evaluation of oscillation mapping in SSc patients

• Thorough evaluation of different binning algorithms and normalisation techniques

• Establishment of an interesting relationship between the oscillation amplitude from the two-key method and the amplitude and phase from the multi-key method

I have some general comments followed by some minor, more specific, comments.

Major Comments:

1. The authors mention that an advantage of the multi-key method is its ability to map both oscillation amplitude and phase, however only oscillation amplitude is analysed here. The RBC oscillation phase is new and relatively unexplored, so it would be interesting to also include these results, i.e. repeatability of phase, comparison between the groups.

2. The paper does a good job of comparing the three mapping methods in terms of their reliability, but some further detail about the differences between the 2-key and multi-key methods is needed. For example, what are the interpretation and implications of the negative amplitude values obtained from the 2-key approach?

3. It is interesting that the reliability of the oscillation maps was not strongly affected by oscillation quality. I can see that for the 2-key method, as long as the maxima and minima can still be identified, noisy oscillations might not cause too much of a problem. However, for the multi-key method, wouldn’t the points chosen by the sliding window become out of sync with each other in cases such as in Figure 4A-C, so that different points in the cardiac cycle are being sampled? How does this impact the phase maps? Do the authors think there is a cut-off for SNR or any other quality metric, or is it okay to perform oscillation mapping regardless of oscillation quality?

Minor Comments:

1. Abstract: add a sentence to summarise how the oscillation mapping method was optimised using healthy volunteer data

2. Abstract, line 25: please include the number of PAH patients included in this study

3. Introduction, line 43: specify that keyhole mapping is possible because of the 3D radial k-space trajectory used

4. Methods, line 87: please rephrase this sentence to mention the physiology that the oscillation measurements reflect

5. Methods, line 91: please explain why you could not compare oscillation mapping to imaging techniques such as DCE-MRI and PREFUL?

6. Methods, ‘Human subjects’: consider renaming this section ‘Participants’ to align with the results section

7. Methods, ‘Binning algorithms’: mention which binning method ref 9 used

8. Methods, line 201 – 202: cut down the amount of words in these equations and replace with symbols where possible

9. Methods, line 222: were you comparing the mean or median values from the maps? Why didn’t you compare oscillation phase between the groups?

10. Results: did you look at the map heterogeneity and how this compares between the groups?

11. Figures 2 and 3: it would perhaps be helpful to include a Bland-Altman plot alongside each correlation plot.

12. Figure 2: why do B and C present the ratios in units of percentage but not A?

13. Figures 6 and 7: I find the colour maps used for the 2-key oscillation amplitude maps a little hard to interpret. Consider using the same colour map as is used for the multi-key method or a map with a larger dynamic range.

14. Figure 9: please explain the outlier in B

15. Discussion: please add an interpretation of the high oscillation amplitudes observed in SSc patients

16. Discussion, line 459: did you look at the correlation of oscillation amplitude with age? In most literature, oscillation amplitude is not correlated with age

**Do you want your identity to be public for this peer review?** For information about this choice, including consent withdrawal, please see our Privacy Policy

Reviewer #1: No

Reviewer #2: No

---

## [Author Response · Author response to Decision Letter 1]

25 Jul 2025

Reviewer comments and our responses are uploaded in the Response to Reviewers document. Thank you for your consideration of our manuscript.

---

## [Decision Letter · Decision Letter 1]

6 Aug 2025

Dear Dr. Niedbalski,

Thank you for submitting your manuscript to PLOS ONE. After careful consideration, we feel that it has merit but does not fully meet PLOS ONE’s publication criteria as it currently stands. Therefore, we invite you to submit a revised version of the manuscript that addresses the points raised during the review process.

**Some minor revisions highlighted by one reviewer are required.**

We look forward to receiving your revised manuscript.

Kind regards,

Vincenzo Lionetti, M.D., PhD

Academic Editor

PLOS ONE

**Journal Requirements:**

Reviewers' comments:

Reviewer's Responses to Questions

**Comments to the Author**

Reviewer #1: All comments have been addressed

Reviewer #2: All comments have been addressed

2. Is the manuscript technically sound, and do the data support the conclusions?

Reviewer #1: Yes

Reviewer #2: Yes

3. Has the statistical analysis been performed appropriately and rigorously?

Reviewer #1: Yes

Reviewer #2: Yes

4. Have the authors made all data underlying the findings in their manuscript fully available?

Reviewer #1: Yes

Reviewer #2: Yes

5. Is the manuscript presented in an intelligible fashion and written in standard English?

Reviewer #1: Yes

Reviewer #2: Yes

**Reviewer #1: ** The authors have done a great job addressing the comments and have made the necessary improvements to the manuscript. I only have one minor comment on the added text that can likely be handled at the editor level.

Comment:

1. Line 538: I don’t follow why the phase maps will be “strongly dependent on the timing of the cardiac cycle with respect to the scan”. To my understanding, the phase is independent of the scan timing as the phase would only be dependent on the timing of the voxel-wise RBC signal maximum relative to the global peak RBC signal maximum. Consider clarifying or removing.

**Reviewer #2: ** I thank the authors for considering and providing detailed responses to all reviewer comments.

**Do you want your identity to be public for this peer review?** For information about this choice, including consent withdrawal, please see our Privacy Policy

Reviewer #1: No

Reviewer #2: No

---

## [Author Response · Author response to Decision Letter 2]

8 Aug 2025

We have responded to Reviewer Comments in the "Response to Reviewers" document attached with our submission. Thank you to the reviewers for their helpful comments.

---

## [Decision Letter · Decision Letter 2]

12 Aug 2025

Optimizing reliability of RBC signal oscillation measures from hyperpolarized 129Xe MRI

PONE-D-25-07475R2

Dear Dr. Niedbalski,

We’re pleased to inform you that your manuscript has been judged scientifically suitable for publication and will be formally accepted for publication once it meets all outstanding technical requirements.

Kind regards,

Vincenzo Lionetti, M.D., PhD

Academic Editor

PLOS ONE

Additional Editor Comments (optional):

Reviewers' comments:

Reviewer's Responses to Questions

**Comments to the Author**

Reviewer #2: All comments have been addressed

2. Is the manuscript technically sound, and do the data support the conclusions?

Reviewer #2: Yes

3. Has the statistical analysis been performed appropriately and rigorously?

Reviewer #2: Yes

4. Have the authors made all data underlying the findings in their manuscript fully available?

Reviewer #2: Yes

5. Is the manuscript presented in an intelligible fashion and written in standard English?

Reviewer #2: Yes

Reviewer #2: (No Response)

**Do you want your identity to be public for this peer review?** For information about this choice, including consent withdrawal, please see our Privacy Policy

Reviewer #2: No

---

## [Editor Report · Acceptance letter]

PONE-D-25-07475R2

PLOS ONE

Dear Dr. Niedbalski,

I'm pleased to inform you that your manuscript has been deemed suitable for publication in PLOS ONE. Congratulations! Your manuscript is now being handed over to our production team.

Kind regards,

on behalf of

Prof. Vincenzo Lionetti

Academic Editor

PLOS ONE